# Compelling Increase in Parvovirus B19 Infections: Analysis of Molecular Diagnostic Trends (2019–2024)

**DOI:** 10.3390/v17040523

**Published:** 2025-04-04

**Authors:** Flora Marzia Liotti, Simona Marchetti, Sara D’Onghia, Lucio Romano, Rosalba Ricci, Maurizio Sanguinetti, Rosaria Santangelo, Brunella Posteraro

**Affiliations:** 1Dipartimento di Scienze di Laboratorio ed Ematologiche, Fondazione Policlinico Universitario A. Gemelli IRCCS, 00168 Rome, Italy; floramarzia.liotti@policlinicogemelli.it (F.M.L.); simona.marchetti@policlinicogemelli.it (S.M.); sara.donghia@policlinicogemelli.it (S.D.); lucio.romano@policlinicogemelli.it (L.R.); rosalba.ricci@policlinicogemelli.it (R.R.); rosaria.santangelo@unicatt.it (R.S.); 2Dipartimento di Scienze Biotecnologiche di Base, Cliniche Intensivologiche e Perioperatorie, Università Cattolica del Sacro Cuore, 00168 Rome, Italy; brunella.posteraro@unicatt.it; 3Unità Operativa “Medicina di Precisione in Microbiologia Clinica”, Direzione Scientifica, Fondazione Policlinico Universitario A. Gemelli IRCCS, 00168 Rome, Italy

**Keywords:** B19V DNA, molecular detection, positivity rate, epidemiological trends, PCR assay, serological assay, laboratory data

## Abstract

Human parvovirus B19 (B19V) follows a well-documented cyclical epidemiology, with peaks occurring every 3–4 years. However, recent reports indicate an unusual resurgence in B19V infections across multiple countries, prompting increased surveillance. This study analyzed molecular diagnostic assay results from 826 unique-patient samples tested for B19V DNA between 2019 and 2024 at a large Italian tertiary-care hospital, covering pre-, during, and post-COVID-19 years. Overall, 80 of 826 patients (9.7%) tested positive for B19V DNA. A significant increase in positivity was observed in 2024 (23.4%), with a peak in May, representing an eightfold rise compared to 2019–2020. Despite this surge, the distribution of positive cases across population categories remained consistent with previous years, with 32 of 80 (40.0%) positive samples from pregnant women and 27 of 80 (33.8%) from hematology/oncology patients. Among 66 B19V DNA-positive patients with available serology, 4 of 66 (6.1%)—all immunocompromised—lacked detectable IgM/IgG despite high B19V DNA levels (7.8 log_10_ IU/mL). These findings highlight the importance of integrating molecular and serological diagnostics, particularly in high-risk populations. Given the potential impact of the COVID-19 pandemic on B19V circulation, continued surveillance is essential to determine whether this resurgence represents a temporary fluctuation or a sustained epidemiological shift.

## 1. Introduction

Human parvovirus B19 (B19V) is a small, single-stranded DNA virus belonging to the *Parvoviridae* family, genus *Erythroparvovirus* [1]. It is the causative agent of erythema infectiosum (fifth disease) in healthy children and arthropathy in healthy adults but can also lead to severe complications in specific populations [2]. Its genome (~5.6 kb) encodes structural (VP1, VP2) and nonstructural (NS1) proteins, which are critical for viral replication and pathogenicity [1]. Among the three recognized B19V genotypes, genotype 1 is by far the most prevalent worldwide [2].

B19V follows a well-documented cyclical epidemiology, characterized by outbreaks occurring every 3 to 4 years, with seasonal peaks in spring and early summer [1]. While often self-limiting, B19V infection poses significant risks to pregnant women [3,4], potentially leading to fetal anemia, nonimmune hydrops, or fetal loss. It is also a serious concern for immunocompromised individuals (e.g., leukemia or other cancers) and patients with red blood cell disorders (e.g., sickle cell disease), in whom persistent anemia or severe inflammatory complications may occur [1]. The conditions share a common challenge: the inability to produce sufficient neutralizing IgG antibodies to the virus. When formed approximately two weeks post-infection, antibodies directed against both VP1 and VP2 proteins are highly effective in eradicating the virus from the bloodstream [2].

In 2024, the European Centre for Disease Prevention and Control (ECDC) issued an alert regarding an unusual increase in B19V cases across Europe [5]. This resurgence has been reported in multiple European countries [6,7,8,9] and in non-European countries, including Israel [10,11] and the U.S. [12]. Diagnostic testing has revealed a marked rise in positivity rates, particularly in the post-COVID-19 pandemic years. In response, the U.S. Centers for Disease Control and Prevention (CDC) released a Health Advisory on 13 August 2024, providing recommendations for healthcare providers, health departments, and the public [13]. Healthcare providers are advised to be aware of increased B19V activity, consider testing individuals at high risk for adverse B19V-related outcomes, and closely monitor high-risk patients for complications.

Since B19V-related symptoms are primarily immune-mediated, serological assays remain the cornerstone of diagnosis in immunocompetent individuals [2]. The detection of IgM and IgG antibodies provides valuable information about recent or past infections. However, in immunocompromised patients, serological responses may be absent, or IgM may persist for extended periods without IgG seroconversion, indicating ongoing infection [1,2]. In such cases, molecular assays (i.e., nucleic acid amplification assays), usually based on real-time polymerase chain reaction (PCR), are crucial for establishing a definitive diagnosis [1,2]. These assays not only confirm active infection but also help differentiate low-level viral persistence from clinically significant viremia and are essential for blood product screening [14].

Despite the availability of both molecular and serological approaches, most epidemiological studies on B19V diagnostic trends have relied primarily on serological data. This may reflect differences in laboratory practices, as molecular testing is not routinely performed in all diagnostic settings.

This study evaluates B19V molecular diagnostic trends over a six-year period (2019–2024) at Fondazione Policlinico Universitario A. Gemelli IRCCS in Rome, Italy, covering pre-, during, and post-COVID-19 periods. Our analysis examines positivity rates stratified by patient condition, highlighting the role of molecular diagnostics in high-risk populations. By assessing B19V circulation patterns and affected clinical categories, this study aims to enhance understanding of the virus’s evolving epidemiology and its implications for public health surveillance.

## 2. Materials and Methods

### 2.1. Data Collection

This retrospective study analyzed real-time PCR-based molecular laboratory-testing results for B19V DNA detection from blood samples collected between January 2019 and December 2024 at Fondazione Policlinico Universitario A. Gemelli IRCCS, a tertiary care hospital in Rome, Italy. Samples were obtained from patients presenting, with B19-specific (e.g., rash or arthralgia) or nonspecific symptoms (e.g., fever, malaise, headache), to the hospital or those hospitalized during the study period and tested by physician request. Only the first positive sample per patient was included in the analysis to ensure data uniqueness. Of the 1016 specimens tested for B19V DNA, 826 represented unique-patient samples, while the remaining 190 were follow-up or monitoring samples from previously tested patients and were therefore excluded. Immunoassay-based serological laboratory-testing results for B19V-specific IgM and IgG antibodies performed on blood samples were concomitantly obtained from the same patients (if available) and tested by physician request.

B19V DNA detection and quantitation were performed using the Conformité Européene (CE)-approved QIAGEN artus Parvo B19 RG PCR Kit (https://www.google.com/search?client=firefox-b-d&channel=entpr&q=artus+Parvo+B19+RG+PCR+Kit+%28QIAGEN; accessed on 28 March 2025) on the Rotor-Gene 3000 system (QIAGEN GmbH, Hilden, Germany), starting from 200 μL of each patient’s whole-blood sample that underwent DNA extraction using the QIAsymphony automated system (QIAGEN GmbH) (https://www.qiagen.com/us/products/discovery-and-translational-research/dna-rna-purification/instruments-equipment/qiasymphony-spas-instruments; accessed on 28 March 2025) [1]. The cycle threshold (Ct) value, representing the number of amplification cycles needed for the PCR signal to reach the threshold of fluorescence, was used to estimate the relative amount of B19V DNA per microliter. The Rotor-Gene 3000 system’s software version 2.3.5 performed this calculation automatically, and the final results were expressed as an international unit (IU)/mL of the original sample volume. As established by the manufacturer, a value greater than 140 IU/mL (2.1 log_10_ IU/mL) was considered indicative of detectable B19V DNA, while values below this threshold were deemed under the assay’s limit of detection. Samples yielding 0 IU/mL indicated no detectable B19V DNA. B19V-specific antibody detection in each patient’s serum sample was performed using the Diasorin LIAISON Biotrin Parvovirus B19 IgG/IgM Plus, a CE-approved chemiluminescent immunoassay intended for the qualitative detection of IgG/IgM antibodies in human blood samples (e.g., serum) (https://www.accessdata.fda.gov/cdrh_docs/pdf22/P220034C.pdf; accessed on 28 March 2025), on the LIAISON XL Analyze (DiaSorin S.p.A., Saluggia, Italy) [1]. Results were reported as a signal-to-cutoff ratio, with values of <0.90 (negative result), 0.90–1.10 (equivocal result), and >1.10 (positive result), as established by the manufacturer.

Patient data such as age, sex, and hospital department/ward were collected.

All data were extracted from the hospital’s clinical microbiology laboratory database.

### 2.2. Data Analysis

Statistical analysis was conducted using SPSS Statistics version 24.0 (IBM Corp., Armonk, NY, USA) and GraphPad Prism version 10.2.3 (GraphPad Software, San Diego, CA, USA). Categorical variables were presented as numbers and proportions, while continuous variables were expressed as medians with interquartile range (IQR), as appropriate.

Monthly positivity rates were calculated to assess annual and seasonal trends. Positivity rates were calculated as the number of positive results divided by the total number of samples tested. Differences in positivity rates across years and between study groups (i.e., patient category) were assessed using the chi-square test.

To assess differences in B19V DNA levels (expressed as log_10_ IU/mL) between samples from patient-category groups or samples from serology-based groups (IgM-positive/IgG-positive, IgM-positive/IgG-negative, IgM-negative/IgG-positive, and IgM-negative/IgG-negative), the Kruskal–Wallis test was used. Pairwise comparisons were performed using the Mann–Whitney U test with Bonferroni correction for multiple comparisons.

Statistical significance was defined by a *p*-value of <0.05.

## 3. Results

We included 826 unique-patient samples, representing 74.0% of the 1016 samples tested for B19V DNA between January 2019 and December 2024. Most samples (675/826, 81.7%) were from adult patients (aged ≥ 18 years), including 98 pregnant women (14.5%). The remaining 151 samples (18.3%) were from pediatric patients. The median age of the 826 patients was 43 years (IQR: 27–63 years). Regarding patient care settings, 669 were hospitalized (inpatients), while 157 were treated as outpatients. Among the 826 patients, 80 tested positive (9.7%) for B19V DNA, with a predominance of female patients (57/80, 71.3%), while the remaining 746 (90.3%) tested negative.

Additionally, although outside the primary scope of this study, 139 samples (4.9% of the unique-patient 2826 samples tested exclusively for B19V serology) were identified during the same period with laboratory-confirmed B19V infection based on the presence of B19V-specific IgM antibodies, indicative of recent infection.

B19V DNA positivity rates (samples tested positive out of total samples tested) remained consistently low from 2019 (4/141, 2.8%) to 2022 (1/94, 1.1%), with sporadic peaks in May 2019 (2/14, 14.3%), April 2020 (1/7, 14.3%), and July 2022 (1/7, 14.3%). No positive cases were recorded in 2021 (0/112, 0.0%) or 2023 (0/71, 0.0%). In contrast, 2024 exhibited the highest positivity rate (72/308, 23.4%), with a peak in May (20/34, 58.8%). Differences in yearly positivity rates were statistically significant for all comparisons (*p* < 0.001). Data from this analysis are summarized in Table 1 and illustrated in Figure 1 and Figure 2.

We analyzed B19V DNA positivity rates from 2019 to 2024, stratifying results by population categories (Figure 3): children (0–14 years), women of reproductive age (20–40 years), and all other patients (excluding the previous groups). While seasonal trends for the entire study population have already been presented in Figure 1, Figure 3 illustrates yearly positivity rate differences among these specific population categories. These differences were statistically significant for all comparisons (*p* < 0.001). Notably, the distribution pattern of B19V DNA positivity among population categories in 2024 closely resembled that observed in 2019, but at proportionally higher positivity rates.

B19V DNA-positive samples (*n* = 80) predominantly originated from specific patient categories, including pregnant women (32/80, 40.0%), hematology/oncology patients (27/80, 33.8%), rheumatology patients (12/80, 15.0%), pediatric/neonatal intensive care unit patients (4/80, 5.0%), transplantation patients (1/80, 1.2%), and neonatology patients (1/80, 1.2%). The remaining 3 samples (3.8%) were from medical/infectious disease patients. These data are summarized in Figure 4, which illustrates the distribution of B19V DNA-positive samples across patient categories. Notably, apart from pregnant women, the first five categories primarily include immunocompromised patients, who are at higher risk of B19V infection and complications. Most B19V DNA-negative samples (421/746, 56.4%) also belonged to these high-risk categories, although they are not shown in Figure 4.

The B19V DNA levels in the 80 positive patients ranged from ≤2.1 to ≥7.8 log_10_ IU/mL, with a median (IQR) of 5.3 (4.5–6.8) log_10_ IU/mL. As shown in Figure 5 and according to the patient categories depicted in Figure 4, the median (IQR) B19V DNA level in pregnant women was 5.4 (5.1–6.0) log_10_ IU/mL. This value differed from those in the hematology/oncology and rheumatology categories, which were 6.2 (3.5–7.8) log_10_ IU/mL and 5.6 (3.8–6.4) log_10_ IU/mL, respectively, but the differences were not statistically significant (*p* > 0.05).

Of the 80 patients, 66 had concomitant B19V IgM/IgG serology results. IgM antibodies were detected in 56 (84.8%) patients, of whom 46 (69.2%) were also positive for IgG antibodies. In six patients (9.1%), only IgG antibodies were detected, while in the remaining four patients (6.1%), all of whom had hematological conditions, neither IgM nor IgG antibodies were present.

Among serology-positive patients (Figure 6), IgM-negative but IgG-positive patients (*n* = 6) had a median (IQR) level of 3.2 (2.3–3.5) log_10_ IU/mL. This value was significantly lower (*p* < 0.001) than the median (IQR) B19V DNA levels in IgM-positive but IgG-negative patients (*n* = 10) and in IgM-positive but IgG-positive patients (*n* = 46), which were 7.1 (6.8–7.4) log_10_ IU/mL and 5.3 (4.8–5.9) log_10_ IU/mL, respectively.

## 4. Discussion

Our analysis of molecular diagnostic laboratory data from 2019 to 2024 at a large Italian tertiary-care hospital revealed a significant increase in B19V DNA positivity in 2024 compared to previous years. The positivity rate in 2024 (23.4%) was approximately eight times higher than in 2019 and 2020, spanning 11 of 12 months and peaking in May. Despite this increase, the distribution of positive cases across population categories—defined by age groups, including women of reproductive age—remained consistent with 2019. Among the 80 B19V DNA-positive samples, 40.0% originated from pregnant women and 33.8% from hematology/oncology patients, reinforcing the impact of B19V on high-risk populations.

This trend aligns with reports from several European countries indicating a resurgence of B19V infections since late 2023, including Italy [15,16]. Mor et al. [11] reported a sharp increase in B19V DNA positivity in Israel, rising from 8.5% in 2020–2022 to 31.0% in 2023. While our study and Mor’s relied on molecular diagnostics, most other reports documenting this resurgence were based on serological data, specifically IgM detection as a marker of recent infection. CDC data highlighted a threefold increase in B19V IgM seropositivity in the U.S. across all age groups, with a notable rise among children aged 5–9 years (from 15% in 2022–2024 to 40% in June 2024) [17]. Similarly, Russcher et al. [8] reported a surge in intrauterine transfusions for B19V-related fetal anemia in Northwestern Europe, emphasizing the severe clinical impact of the outbreak. These findings collectively suggest a widespread increase in B19V circulation, affecting both pediatric and high-risk adult patients, underscoring the need for enhanced molecular and serological surveillance strategies.

The integration of molecular and serological diagnostics is crucial for characterizing B19V infection patterns, as demonstrated by Veyrenche et al. [18]. Their study, covering primary B19V infection cases from 2012 to 2024—including the 2023–2024 European outbreak—identified four distinct virological profiles based on combined B19V DNA quantification and IgM/IgG serology, aligning with the known pathophysiology and temporal sequence of infection [19]. Notably, cases of chronic anemia exhibited low IgM/IgG levels alongside a high B19V DNA load, with a median of 7.7 log_10_ IU/mL. Similarly, our study, which analyzed single-patient blood samples as a proxy for primary infection and inferred clinical conditions from hospital ward data, mirrors the approach of Veyrenche et al. [18]. Among the 80 B19V DNA-positive patients, 66 had both molecular and serological data available. Notably, four patients, all likely immunocompromised, had the highest B19V DNA levels (7.8 log_10_ IU/mL) but lacked detectable IgM/IgG antibodies. This contrasts with the virological profile observed in erythema infectiosum, where children typically present with high IgM/IgG titers and lower viral loads—a profile not assessed in our study.

Although samples from pregnant women accounted for only 14.5% of those tested in our study, they represented 40.0% (32/80) of B19V DNA-positive cases. Notably, all 32 positive pregnant women also underwent serological testing, reinforcing the utility of a combined molecular-serology diagnostic approach [4]. Diagnosing B19V in pregnancy is challenging, as the infection is often asymptomatic (30–50% of cases) or presents with nonspecific febrile illness, rash, or arthropathy (30–40% of cases) [8]. In nonimmune pregnant women, vertical transmission can lead to severe complications such as fetal anemia and hydrops, which may require intrauterine transfusion to mitigate fetal mortality risk [4]. Consistently, Russcher et al. [8] reported that, by early 2024, intrauterine transfusions for B19V-related fetal anemia had already exceeded the annual maximum recorded since 2010 in Northwestern Europe (Leiden, Netherlands; Leuven, Belgium; Paris, France). While a low threshold for serological B19V testing is recommended [17], integrating antibody detection with B19V DNA testing improves result interpretation, particularly since pregnant women may present at different stages of infection.

The seasonal trends observed in our study, with infection peaks in specific months, align with the established seasonality of B19V, which typically exhibits annual peaks in late spring. However, recent studies [7,8] have reported disruptions in this pattern, particularly since 2014, with a decline in B19V cases during and immediately after the COVID-19 pandemic. Our analysis, spanning the pre-, during-, and post-pandemic periods (2019–2024), reflects this shift, showing an absence of detected cases in 2021 and 2023, followed by a marked resurgence in 2024, peaking in May. These findings support the hypothesis that changes in population susceptibility and transmission dynamics may have influenced B19V circulation in recent years. Further investigation is needed to clarify these factors and refine surveillance strategies.

This study has some limitations. By focusing exclusively on molecular diagnostic trends over the study years, it provides a restricted view of B19V epidemiology. While our study offers a direct assessment of B19V DNA in the patient’s blood through real-time PCR-based detection, it does not account for infections identified solely through serology. Notably, among the 2826 single-patient samples tested exclusively for B19V serology, 139 (4.9%) had IgM-positive results, indicating recent infection. This suggests that relying only on molecular data may underestimate the true burden of B19V infections. We also acknowledge that the absence of B19V-specific antibodies in some patients with viremia could reflect the acute phase of infection rather than an underlying immunosuppressed state. This limitation highlights the importance of interpreting molecular and serological data together, particularly in high-risk populations. As our study was conducted retrospectively and based on molecular testing data from a single center, we were unable to systematically collect clinical information, including pregnancy outcomes, which limits the ability to fully contextualize B19V epidemiology. Future studies should aim to integrate comprehensive clinical and outcome data to provide a more robust understanding of B19V circulation and its clinical impact. Nevertheless, our study complements serological surveillance by providing valuable insights into B19V circulation in high-risk populations. These findings highlight the need for integrated screening strategies that combine molecular and serological diagnostics, ensuring a more comprehensive understanding of B19V circulation. Increasing healthcare professionals’ awareness of B19V infection patterns and diagnostic challenges could further improve timely detection and patient management.

## 5. Conclusions

The significant rise in B19V DNA-positive cases identified in our study during 2024 underscores the need for heightened clinical vigilance, particularly among high-risk populations. The seasonal peaks in spring and early summer reinforce the importance of routine diagnostic testing during these periods. Given that recent epidemiological trends may have been influenced by the COVID-19 pandemic, continued surveillance is essential to determine whether the increase represents a long-term shift in B19V circulation. Strengthening molecular and serological monitoring strategies will be key to providing a clearer epidemiological picture. These findings support the need for an integrated and multidisciplinary approach to mitigate public health risks and effectively respond to the ongoing ECDC alert.

## Figures and Tables

**Figure 1 viruses-17-00523-f001:**
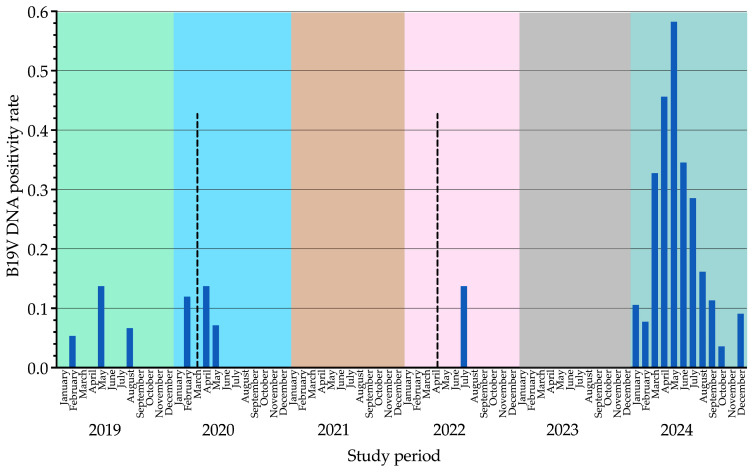
Seasonal trends of human parvovirus B19 DNA detection in blood samples from January 2019 to December 2024. Blue bars represent the monthly positivity rate, calculated as the number of positive samples divided by the total number of samples tested. Dotted lines highlight the COVID-19-related years (2020–2022) within the six-year study period. A different background color is used to distinguish each year, with month labels aligned for clarity. B19V, parvovirus B19.

**Figure 2 viruses-17-00523-f002:**
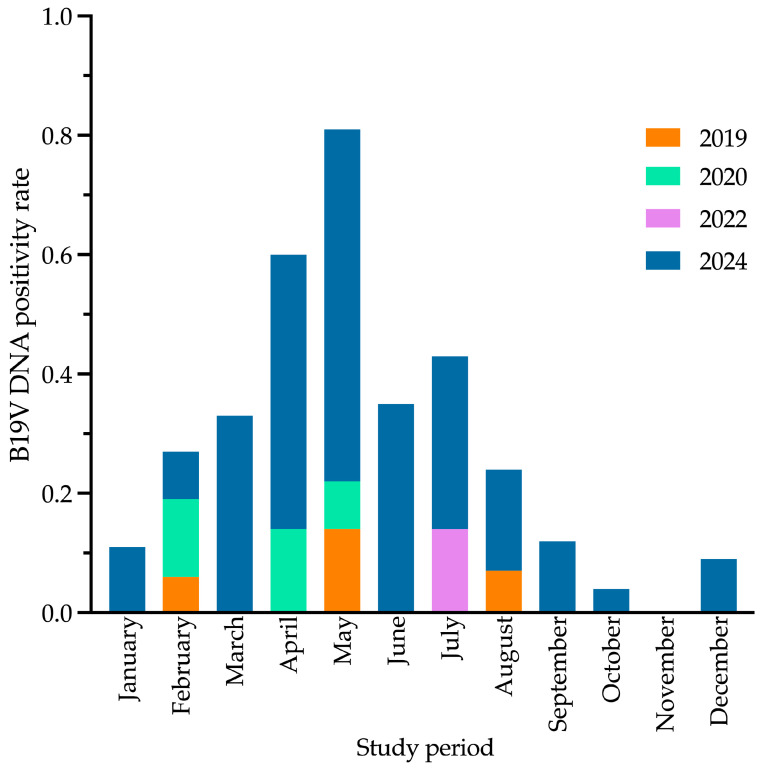
Cumulative monthly positivity rates of human parvovirus B19 DNA detection in blood samples across study years. Bars represent the proportion of positive samples per month, aggregated by year. Each color corresponds to a different year, highlighting the variation in testing and positivity rates over time. The year 2024 spans almost the entire monthly range, while other years are represented by only one to a few months. No positive samples were detected in 2021 and 2023, as detailed in Table 1. For details on positivity rate calculation, see Figure 1. B19V, parvovirus B19.

**Figure 3 viruses-17-00523-f003:**
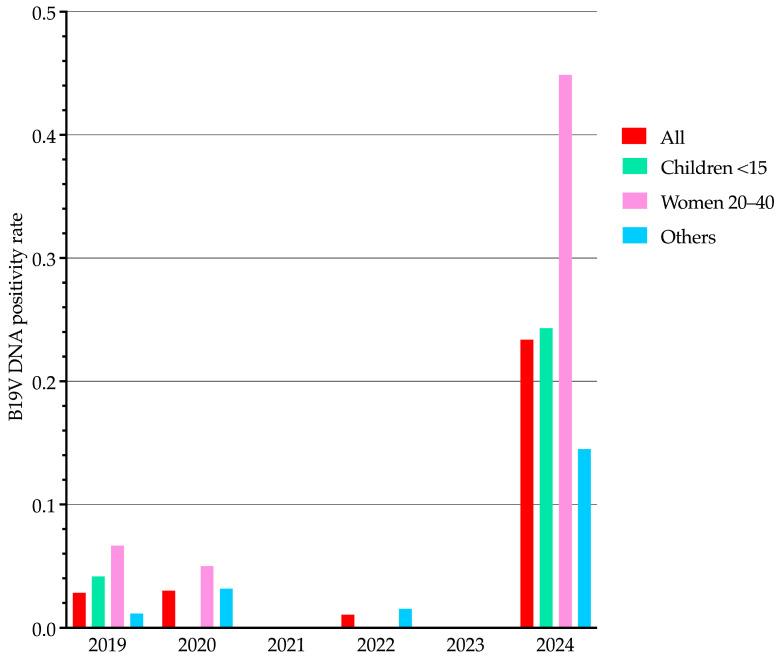
Trends of human parvovirus B19 DNA detection in blood samples from January 2019 to December 2024, stratified by population category. Bars represent the proportion of positive samples per year, with each color corresponding to a different category as indicated in the legend. “All” refers to the entire study population. For details on positivity rate calculation, see Figure 1. B19V, parvovirus B19.

**Figure 4 viruses-17-00523-f004:**
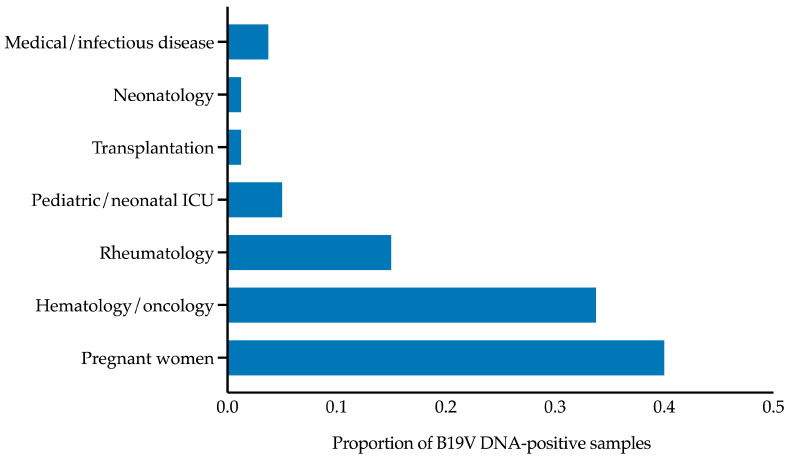
Single-patient samples that tested positive for human parvovirus B19 DNA from January 2019 to December 2024 (*n* = 80), stratified by patient category. Pregnant women (*n* = 32) represented the largest group, followed by hematology/oncology patients (*n* = 27). B19V, parvovirus B19; ICU, intensive care unit.

**Figure 5 viruses-17-00523-f005:**
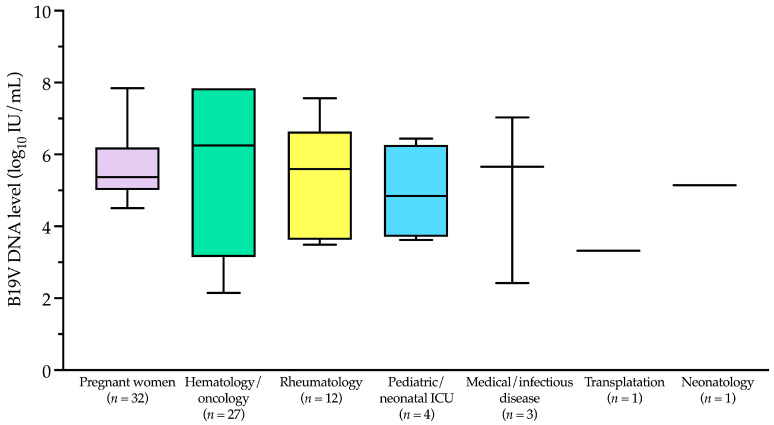
Human parvovirus B19 DNA levels among patient categories’ samples tested from January 2019 to December 2024. Boxplots with whiskers represent the distribution of B19V DNA levels across different categories, with the central line indicating the median, the box representing the interquartile range (IQR), and the whiskers extending to the minimum and maximum values within 1.5 times the IQR. No statistical significance was observed across all comparisons. B19V, parvovirus B19; ICU, intensive care unit; IU, international unit.

**Figure 6 viruses-17-00523-f006:**
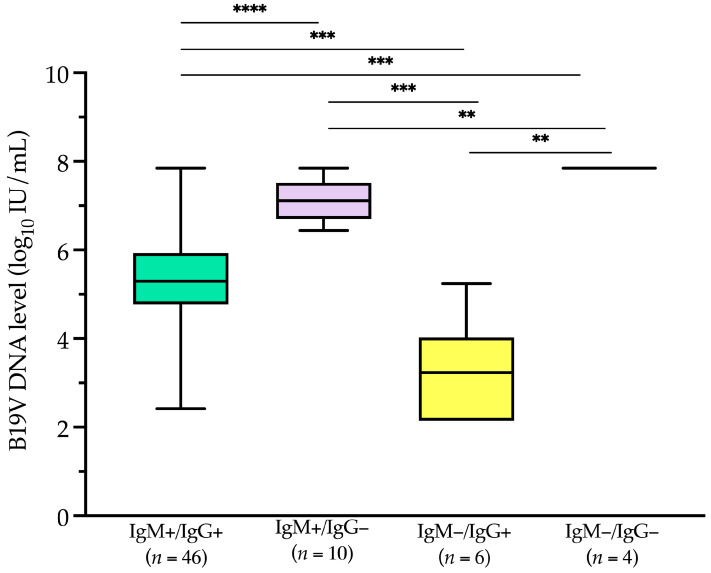
Human parvovirus B19 DNA levels among serology-based categories’ samples tested from January 2019 to December 2024. Boxplots with whiskers represent the distribution of B19V DNA levels across different categories, with the central line indicating the median, the box representing the interquartile range (IQR), and the whiskers extending to the minimum and maximum values within 1.5 times the IQR. Asterisks indicate statistically significant differences (****, *p* < 0.0001; ***, *p* < 0.001; **, *p* < 0.01). B19V, parvovirus B19; IgM+, immunoglobulin M-positive; IgM−, immunoglobulin M-negative; IgG+, immunoglobulin G-positive; IgG−, immunoglobulin G-negative; IU, international unit.

**Table 1 viruses-17-00523-t001:** B19V DNA testing results for 826 blood samples included in the study.

	No. of Samples Tested Positive/No. of Samples Tested in Total (%), as Stratified by Year ^1^					
Months by Year	2019	2020	2021	2022	2023	2024
January	0/15 (0.0)	0/11 (0.0)	0/8 (0.0)	0/10 (0.0)	0/11 (0.0)	1/9 (11.1)
February	1/17 (5.9)	1/8 (12.5)	0/2 (0.0)	0/9 (0.0)	0/5 (0.0)	1/12 (8.3)
March	0/18 (0.0)	0/3 (0.0)	0/15 (0.0)	0/7 (0.0)	0/6 (0.0)	2/6 (33.3)
April	0/7 (0.0)	1/7 (14.3)	0/10 (0.0)	0/9 (0.0)	0/5 (0.0)	6/13 (46.2)
May	2/14 (14.3)	1/13 (7.7)	0/10 (0.0)	0/13 (0.0)	0/5 (0.0)	20/34 (58.8)
June	0/17 (0.0)	0/18 (0.0)	0/11 (0.0)	0/9 (0.0)	0/7 (0.0)	13/37 (35.1)
July	0/8 (0.0)	0/9 (0.0)	0/7 (0.0)	1/7 (14.3)	0/4 (0.0)	16/55 (29.1)
August	1/15 (6.7)	0/2 (0.0)	0/7 (0.0)	0/8 (0.0)	0/6 (0.0)	5/30 (16.7)
September	0/10 (0.0)	0/7 (0.0)	0/6 (0.0)	0/5 (0.0)	0/6 (0.0)	5/42 (11.9)
October	0/6 (0.0)	0/10 (0.0)	0/10 (0.0)	0/10 (0.0)	0/7 (0.0)	1/28 (3.6)
November	0/9 (0.0)	0/7 (0.0)	0/12 (0.0)	0/4 (0.0)	0/4 (0.0)	0/20 (0.0)
December	0/5 (0.0)	0/5 (0.0)	0/14 (0.0)	0/3 (0.0)	0/5 (0.0)	2/22 (9.1)
Total months	4/141 (2.8)	3/100 (3.0)	0/112 (0.0)	1/94 (1.1)	0/71 (0.0)	72/308 (23.4)

^1^ All samples were from unique patients who were either hospitalized (inpatients) or treated as outpatients. Positive samples had B19V DNA levels that ranged from ≤2.1 to ≥7.8 log_10_ IU per milliliter.

## Data Availability

Deidentified data supporting the findings of this study are available upon reasonable request from the corresponding author.

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
