# Peer review of "Compelling Increase in Parvovirus B19 Infections: Analysis of Molecular Diagnostic Trends (2019–2024)"

_viruses, 2025, doi:10.3390/v17040523_

Round 1

Reviewer 1 Report

Comments and Suggestions for Authors

The article by Liotti and colleagues shows in a very compelling and elegant manner the increase in circulation of Parvovirus B19 in Rome and aligns with other reports at the global label showing a similar trend in circulation, specially after the SARS CoV-2 pandemic.  In general is a clear article and I only have minor comments in format and suggestions to improve the clarity of the figures.

Lines 52-53.  The phrase is little confusing. I do understand that the antibodies generated against VP1 and/or VP2 proteins, that might be generated two weeks post infection, are capable to eradicate the infection, is just the way it is written what is confusing.

Figure 1. I suggest that the doted lines that enclosed the pandemic months do not overlap with the bar. Also please align the bar with the month. And If posible label each year and mark the month, maybe add a different color in the background from each january-november.

Figure 5. If there is no significant difference  in any of the comparisons, it is not necesary to add so much labels on top, just mention it in the legend.

Author Response

Comments 1: 

The article by Liotti and colleagues shows in a very compelling and elegant manner the increase in circulation of Parvovirus B19 in Rome and aligns with other reports at the global label showing a similar trend in circulation, especially after the SARS CoV-2 pandemic. In general, this is a clear article, and I only have minor comments in format and suggestions to improve the clarity of the figures.

Lines 52–53. The phrase is a little confusing. I do understand that the antibodies generated against VP1 and/or VP2 proteins, that might be generated two weeks post infection, are capable of eradicating the infection, is just the way it is written what is confusing.

Response 1: We thank the reviewer for pointing out this issue. We have revised the phrase to improve clarity as follows: When formed approximately two weeks post-infection, antibodies directed against both VP1 and VP2 proteins are highly effective in eradicating the virus from the bloodstream [2]. See text in the revised version of the manuscript.

Comments 2: Figure 1. I suggest that the dotted lines that enclosed the pandemic months do not overlap with the bar. Also please align the bar with the month. And if possible label each year and mark the month, maybe add a different color in the background from each January November.

Response 2:  We appreciate the valuable suggestions regarding Figure 1. We have modified the figure as recommended, ensuring that the dotted lines representing the pandemic months do not overlap with the bars. We have also aligned the bars with the months and added year labels. Additionally, we have included a subtle background color difference between January and December to enhance clarity. See Figure 1 in the revised version of the manuscript.

Comments 3: If there is no significant difference in any of the comparisons, it is not necessary to add so many labels on top, just mention it in the legend.

Response 3: We thank the reviewer for the comment. We have revised Figure 5 to minimize the number of labels on top, retaining only the most relevant information. A statement has been added to the figure legend to clarify the absence of statistically significant differences where applicable. See Figure 5 in the revised version of the manuscript.

Reviewer 2 Report

Comments and Suggestions for Authors

This epidemiological study is of considerable help in understanding the prevalence of B19 in the local area from 2019 to 2024, but there are still some deficiencies in the current manuscript.

1.As the significance of DNA testing and antibody testing for B19 is different, I suggest deleting the content of seroepidemiology. However, the research results of local seroepidemiology can be used as a reference for the detection rate of viremia. The author's viewpoint that some patients with possible immunosuppression may have viremia but negative antibodies might be incorrect. It is possible that during the acute phase of infection, IgM and IgG have not yet been produced.

2.As this study is based on data from a single center, theoretically, it can be analyzed in combination with clinical information and the results of blood B19-DNA testing, especially the clinical conditions of pregnant women and their offspring, at least whether there are relevant clinical diagnoses.

3.I suggest reducing the number of pictures (Fig2-6)and replacing them with textual descriptions.

4. Some puzzling data need to be explained, such as why 826 cases (specimens) were selected from 1,016 specimens?

Author Response

Comments 1: 

This epidemiological study is of considerable help in understanding the prevalence of B19 in the local area from 2019 to 2024, but there are still some deficiencies in the current manuscript.

  1. As the significance of DNA testing and antibody testing for B19 is different, I suggest deleting the content of seroepidemiology. However, the research results of local seroepidemiology can be used as a reference for the detection rate of viremia. The author’s viewpoint that some patients with possible immunosuppression may have viremia but negative antibodies might be incorrect. It is possible that during the acute phase of infection, IgM and Ig G have not yet been produced.

Response 1: We thank the reviewer for the insightful comment. We understand the concern regarding the inclusion of serological data. We believe that molecular and serological data complement each other, particularly in high-risk populations such as pregnant women and immunocompromised patients. To address the reviewer’s suggestion, we have added a statement to the limitations section acknowledging that the absence of antibodies in some patients could be related to the acute phase of infection rather than immunosuppression. We opted to retain the serological data as it provides valuable insights into the diagnostic challenges associated with B19V infections, especially in settings where rapid differentiation between acute and chronic cases is essential. See text in the revised version of the manuscript.

Comments 2: As this study is based on data from a single center, theoretically, it can be analyzed in combination with clinical information and the results of blood B19-DNA testing, especially the clinical conditions of pregnant women and their offspring, at least whether there are relevant clinical diagnoses.

Response 2: We agree with the reviewer that integrating clinical data would enhance the robustness of the analysis. However, due to the retrospective nature of our study and the limited availability of clinical information, including pregnancy outcomes, such data were not systematically collected. We have acknowledged this limitation in the discussion and have recommended that future studies include more comprehensive clinical and outcome data to better contextualize B19V epidemiology. See text in the revised version of the manuscript.

Comments 3: I suggest reducing the number of pictures (Fig 2-6)and replacing them with textual descriptions.

Response 3: We thank the reviewer for the suggestion. However, we believe that each figure provides unique and valuable information that enhances the clarity and impact of the study. Moreover, Reviewer 1 has explicitly requested improvements to some of these figures rather than their removal, highlighting their relevance to the manuscript. To address both reviewers' concerns, we have carefully revised the figures to enhance clarity and improve their visual presentation, while maintaining the full set to ensure comprehensive data visualization.

Comments 4: Some puzzling data need to be explained, such as why 826 cases (specimens) were selected from 1,016 specimens?

Response 4: We thank the reviewer for pointing out this aspect. The discrepancy arises because we included only unique-patient samples in the study. Among the 1,016 specimens tested for B19V DNA, 826 represented unique-patient samples, while the remaining 190 were follow-up or monitoring samples from previously tested patients. This approach was taken to avoid duplication and ensure data consistency, and the clarification has now been added to the Materials and Methods section. See text in the revised version of the manuscript.